# Carboxymethylcytosine is a natural base modification and a handle for bacteriophage DNA hypermodification

Qiaoyu Yang[1,2,3,10], Lin Zhang [4,10], Yantao Liang [5,10], Haoyu Ma[6], Lifu Song[7], Lin Luo[5], Jason Tan[8], Yiling Hu[1], Kailiang Ma[1], Yiwei Chen[1], Yang Tong[1], Chuyuan Zhang[1], Suwen Zhao [6] ✉, Min Wang [5] ✉, Liang Zhang [4] ✉, Yifeng Wei [8] ✉ & Yan Zhang [1,2,3,9] ✉

Bacteriophages possess a wide array of DNA modifications, with many acting as molecular camouflage to evade host immune defenses. Sequence databases contain numerous bacteriophage enzymes of unknown function, with some potentially involved in yet to be identified DNA modifications. Here we report the discovery of a DNA cytosine C5-carboxymethyltransferase (CmoX) in *Synechococcus* phage S-B43, which catalyzes the formation of a 5-carboxymethylcytosine (5cxmC), previously reported as an unnatural DNA modification formed by an engineered cytosine methyltransferase. The carboxy-S-adenosyl-L-methionine (Cx-SAM) cofactor required by CmoX is provided by a phage-encoded Cx-SAM synthase (CmoA), a homolog of the bacterial CmoA involved in tRNA modification. A crystal structure of CmoX in complex with Cx-SAM revealed the basis for its substrate selectivity, involving a key Arg residue interacting with the substrate carboxy group. In addition, we characterize a phage-encoded ATP-dependent amide ligase, CmoY that catalyzes the formation of 5cxmC-glycine amide. CmoA is present in many bacteriophage genomes, typically alongside CmoX and homologs of CmoY, suggesting that 5cxmC modification is a widespread naturally occurring DNA modification serving as a handle for further hypermodifications in bacteriophages. Our study underscores the ability of bacteriophages to repurpose RNA modification enzymes to expand their repertoire of DNA modifications.

DNA modifications play important roles in all domains of life, including eukarya, bacteria, archaea, and viruses[1]. In vertebrate epigenetics, 5-methylcytosine (5mC) modifications within CpG motifs function in maintaining cell identity, regulating gene expression and chromatin structure, as well as ensuring genome stability by suppressing transposable elements[2,3]. In bacterial restriction-modification (RM) systems, 5mC, N6-methyladenine (6mA) and N4-methylcytosine (4mC) added by modification methyltransferases serve to distinguish self and foreign DNA, protecting bacterial genomes from their own restriction enzymes[1]. In comparison, tailed bacteriophages in the class Caudoviricetes possess far more diverse and complex DNA modifications that aid in evading bacterial RM systems, likely arising from an evolutionary arms race between phages and their hosts[4,5]. The study of phage DNA modifications has broadened our understanding of DNA structure and of nucleotide biosynthetic pathways, with promising applications in various DNA-based technologies. The sheer abundance and variety of bacteriophages in nature suggests that many phage DNA modifications and DNA modifying enzymes remain to be discovered.

A full list of affiliations appears at the end of the paper. ✉e-mail: zhaosw@shanghaitech.edu.cn; mingwang@ouc.edu.cn; liangzhang2014@sjtu.edu.cn; weiyf@a-star.edu.sg; yan.zhang@tju.edu.cn

Phage DNA modifications generally occur either at the nucleotide level prior to incorporation by DNA polymerase, or at the nucleic acid level after the DNA has already been synthesized. An example of nucleotide level modification is 2-amino-dA (Z), which involves conversion of dGMP to dZMP by a phage-encoded succinyl-2-amino-dAMP synthetase (PurZ/PurZ0)[6] and bacterial adenylosuccinate lyase (PurB), followed by phosphorylation to dZTP and incorporation by phage DNA polymerase[7–9]. An example of nucleic acid level modifications is the substitution of guanine by 7-deazaguanine in certain phage genomes, catalyzed by DNA-guanine transglycosylase (DpdA)[10]. This DNA modification system is homologous to the 7-deazaguanine tRNA modification system[10], involving synthesis of the 7-deazaguanine base, and incorporation into tRNA via tRNA-guanine transglycosylase (Tgt), a distant homolog of DpdA[11,12]. In certain phages, the 7-deazaguanine base undergoes further modifications, enhancing structural diversity[13].

Compared to DNA, RNA exhibits a far greater variety of modifications, with broad effects on folding, structural stability, intermolecular interactions, splicing, gene expression, and immunogenicity[14]. Structural diversity is highest in tRNA, with over 100 distinct modifications identified[15], mostly at the 5' wobble position of the anticodon loop[16]. We decided to examine whether any of these RNA modifications may have been repurposed by phages. We noticed a *cmoA* gene, involved in 5-carboxymethoxyuridine (cmo5U) modification of Gram-negative bacteria tRNA[17], is present in certain phages. The carboxymethyl moiety originates from the unusual cofactor Cx-SAM, synthesized by the SAM carboxylase CmoA using prephenate as a co-substrate and involving a unique sulfonium ylide intermediate[18]. In *E. coli*, the $O_2$-dependent TrhO or prephenate-dependent TrhP catalyzes the formation 5-hydroxyuridine (ho5U) in tRNA[17]. The methyltransferase homolog CmoB then catalyzes carboxymethyl transfer from Cx-SAM to ho5U, yielding mature cmo5U-modified tRNA[18]. In some cases, the carboxylate group is further methylated by the SAM-dependent methyltransferase CmoM, forming 5-methoxycarbonylmethoxyuridine[19].

Here, we report the discovery of an unprecedented DNA modification system, involving synthesis of Cx-SAM by the phage CmoA, and DNA cytosine carboxymethylation by a homolog of DNA cytosine-5-methyltransferase (C5-MTase), which we name CmoX. The carboxymethyl group serves as a handle for subsequent hypermodification through ATP-dependent ligation to glycine catalyzed by an ATP-dependent ligase (CmoY). We present the crystal structure of CmoX, and examine its distribution across diverse bacteriophages, supporting the existence of a widespread modification-hypermodification system.

## Results
### Identification of CmoA homologs in bacteriophages
While searching for homologs of RNA modification enzymes in phage, we observed the presence of CmoA homologs in several phages, including *Synechococcus* phage S-B43[20] (Fig. 1a), which lacks a homolog of *E. coli* CmoB. Instead, the CmoA genome neighborhood contains a homolog of C5-MTase (which we designate CmoX) (Fig. 1a), suggesting that this enzyme may in fact catalyze DNA cytosine-C5-carboxymethylation (Fig. 1b). This reaction has previously been observed only in an engineered variant of the methyltransferase M.MpeI and the 5cxmC product was called unnatural base[21]. In addition, the genome neighborhoods also often contain a homolog of asparagine synthetase (which we designate CmoY) (Fig. 1a), suggesting further hypermodification by ATP-dependent ligation to an unknown amine (which we later identified as glycine, Fig. 1b). Sequence comparisons of these phage enzymes with their related *E. coli* homologs are summarized in Supplementary Table 1. To investigate this biochemical hypothesis, the enzymes were recombinantly produced in *E. coli* and purified (Supplementary Fig. 1).

### *Sp*CmoA is a highly active Cx-SAM synthase
*Sp*CmoA and *Ec*CmoA as a positive control were separately incubated with SAM and prephenate for 1 h, followed by precipitation of the proteins and analysis of the reaction products by HPLC and LC−MS (Fig. 2). The substrate SAM elutes with retention time of 26.57 min and positive ion $m/z$ of 399.1. Incubation with either *Sp*CmoA or *Ec*CmoA led to a decrease in the SAM peak, and the appearance of a product peak with retention time of 23.48 min and a positive ion $m/z$ of 443.0, corresponding to Cx-SAM, demonstrating that *Sp*CmoA is indeed a Cx-SAM synthase. A significantly higher product yield was observed for *Sp*CmoA, and subsequent assays determined $k_{cat}$ values of 2.33 $s^{-1}$ for *Sp*CmoA and 0.23 $s^{-1}$ for *Ec*CmoA (Supplementary Fig. 2). The *Sp*CmoA reaction mixture was thus used as a source of Cx-SAM for subsequent biochemical assays.

### *Sp*CmoX is a DNA cytosine C5-carboxymethyltransferase
SEC analysis indicated that recombinant *Sp*CmoX exists as a monomer in solution (Supplementary Fig. 3). To assay for carboxymethyltransferase activity, *Sp*CmoX was incubated with a GC-rich 110-bp dsDNA fragment, and the *Sp*CmoA reaction mixture as a source of Cx-SAM, followed by purification and digestion of the dsDNA product to nucleosides. LC−MS analysis revealed the presence of a product peak with a positive ion $m/z$ of 286.1 (Fig. 3a, b) and a MS/MS fragmentation pattern consistent with 5-carboxymethyl-2'-deoxycytidine (5cxmdC) (Supplementary Fig. 4). To investigate the effect of carboxymethylation on restriction enzyme digestion, HhaI (cleavage sequence GCG^C) was incubated with purified carboxymethylated dsDNA, or unmodified dsDNA as a control. Analysis by agarose gel electrophoresis revealed that carboxymethylation confers resistance to HhaI digestion (Fig. 3c).

We next attempted in vivo reconstitution of the CmoA-CmoX system in *E. coli* BL21(DE3) to determine whether it could modify plasmid DNA. Since *E. coli* natively expresses CmoA to produce the carboxymethyl donor Cx-SAM, we expressed only *SpcmoX*, from a modified pET28a (+)-HT vector. To investigate the requirement for CmoA, a ΔcmoA strain was constructed using a CRISPR-aided homologous recombination method, confirmed by colony PCR. The plasmid HT-*SpcmoX* was transformed into the wild-type and ΔcmoA strains, followed by growth and induction of *SpcmoX* with IPTG. The plasmids were then extracted and digested to nucleosides, followed by LC−MS analysis (Fig. 3d,e). A 5cxmdC peak with retention time of 3.30 min and positive ion $m/z$ of 286.1 was only observed for wild-type but not the ΔcmoA strain, consistent with the ability of the bacterial CmoA to supply Cx-SAM for the carboxymethylation reaction.

### Nanopore sequencing identifies a GC motif required for modification
To investigate the presence of modified bases in the genomic DNA of *Synechococcus* phage S-B43, the DNA was extracted, followed by sequencing by Oxford Nanopore technology. Analysis of the sequencing data using Tombo 1.5.1 revealed a cytosine with a high likelihood of modification (Fig. 4a), although its chemical structure remained undetermined. Sequences surrounding modified cytosine were analyzed using the MEME suit and visualized using WebLogo (Fig. 4b), revealing that the modification occurs within a highly conserved GC motif.

To further verify the sequence specificity of *Sp*CmoX, a series of palindromic dsDNA substrates containing GC motifs were incubated with *Sp*CmoX, then digested and analyzed by LC−MS to detect the presence of 5cxmdC. Controls included no *Sp*CmoX and dsDNA substrates that do not contain any GC motif. Enzyme activity was detected only in assays with dsDNA substrates containing GC motifs, confirming that this motif is essential for recognition by *Sp*CmoX (Fig. 4c, d).

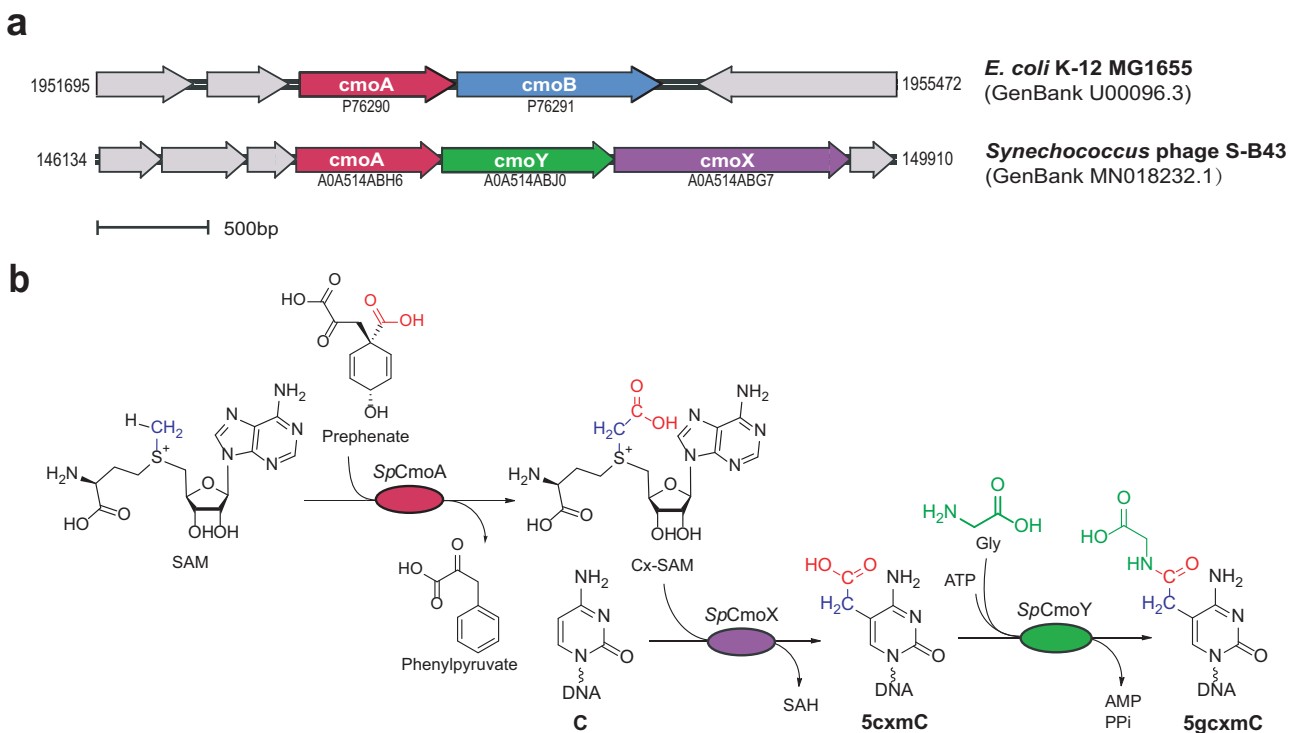

**Fig. 1 | A putative 5-glycylcarboxymethylcytosine (5gcxmC) synthesis pathway. a** Gene clusters containing *cmoA* in *E. coli* and *Synechococcus* phage S-B43. UniProt accession numbers for coded proteins are indicated. **b** Proposed pathway for 5gcxmC synthesis.

## Crystal structure of *Sp*CmoX in complex with Cx-SAM

To gain insights into the catalytic mechanism and substrate selectivity of CmoX, the crystal structure of *Sp*CmoX in complex with the substrate Cx-SAM was determined to 1.9 Å resolution (Fig. 5, Supplementary Table 4). Like other C5-MTases, *Sp*CmoX adopts a $\beta/\alpha$-barrel fold (Fig. 5a). Key interactions between Cx-SAM and the active site residues of *Sp*CmoX are detailed in Fig. 5b. The Cx-SAM adenine ring is stabilized by hydrogen bonding with Asp-69 and stacking interactions with aromatic residues Phe-25 and Phe-48, while the 2'-OH and 3'-OH of ribose form hydrogen bond interactions with Glu-47. A model of the *Sp*CmoX ternary complex was constructed using an alignment with the crystal structure of HhaIM in complex with S-adenosyl-L-homocysteine (SAH) and a dsDNA (PDB: 1MHT, Fig. 5c). The Cys-81 nucleophile in HhaIM forms a covalent bond with cytosine C6, and aligns with Cys-98 in *Sp*CmoX. Notably, Asn-304 in HhaIM is replaced with Arg-350 in *Sp*CmoX, and is in direct interaction with the Cx-SAM carboxymethyl group. The interaction of the Cx-SAM carboxymethyl group with a specific Arg residue was previously proposed to contribute to substrate recognition by CmoB from *Vibrio vulnificus*[22]. We also tested whether the *Sp*CmoX mutant R350N could accept SAM or Cx-SAM as substrates. The results showed that *Sp*CmoX (R350N) could not accept either, while the wild-type could only accept Cx-SAM (Supplementary Fig. 5). The crystal structure of *Sp*CmoX is consistent with a catalytic mechanism for cytosine-5-carboxymethylation that is analogous to related SAM-dependent methyltransferases[4]. In this pathway, nucleophilic attack of Cys-98 on the C6 position of cytosine forms a covalent enzyme-DNA carbanion intermediate, followed by transfer of the Cx-SAM carboxymethyl group onto the cytosine C5 carbanion, forming SAH. Finally, β-elimination of Cys-98 is facilitated by the deprotonation of C5, releasing the carboxymethylated DNA product (Fig. 5d).

## *Sp*CmoY ligates DNA 5cxmC to glycine

*Sp*CmoA and *Sp*CmoX are located adjacent to a gene encoding a homolog of asparagine synthetase, an ATP-dependent amide ligase, which we designate *Sp*CmoY. The genomic proximity of *Sp*CmoY with CmoX suggests that *Sp*CmoY may catalyze ATP-dependent ligation of the carboxymethyl group to an unidentified amine substrate. To test this hypothesis, *E. coli* BL21(DE3) strains harboring either the plasmid HT-*SpcmoX* (expressing *Sp*CmoX, Fig. 6a) or the dual-expression plasmid HT-*SpcmoX*-*SpcmoY* (co-expressing *Sp*CmoX and *Sp*CmoY, Fig. 6b) were constructed. The genomic DNAs from both strains were extracted and digested to nucleosides, followed by LC–MS analysis (Fig. 6c, d). Expression of *Sp*CmoX resulted in a 5cxmdC peak (retention time of 3.42 min and positive ion *m/z* of 286.1), consistent with previous observations above. Strikingly, co-expression of *Sp*CmoX and *Sp*CmoY abolished the 5cxmdC peak and generated a new peak (retention time of 3.36 min and *m/z* 343.1) consistent with glycine ligation to form 5-glycylcarboxymethyl-2'-deoxycytidine (5gcxmdC) (Fig. 6e, f), as corroborated by MS/MS fragmentation (Supplementary Fig. 6).

To further validate this modification in vitro, 5cxmC-modified genomic DNA from the HT-*SpcmoX* transformant strain was incubated with purified *Sp*CmoY in the presence of ATP and glycine. Workup and LC–MS analysis confirmed the formation of 5gcxmC (*m/z* 343.2 → 227.1), which was absent in negative controls omitting ATP, enzyme, or glycine (Fig. 6g). The reaction was repeated with a synthetic 110-bp PCR-amplified dsDNA substrate modified first with *Sp*CmoX and then *Sp*CmoY, further confirming the formation of 5gcxmC (Supplementary Fig. 7). In analogy to related ATP-dependent ligases, we propose a catalytic mechanism, involving adenylation of the carboxyl group of 5cxmC, followed by nucleophilic attack by glycine, forming the amide bond in the modified base 5gcxmC (Fig. 6h).

## Analysis of the *Synechococcus* phage S-B43 genomic DNA

We initially investigated the 5gcxmC modification in the recombinant system above due to its experimental simplicity compared to the native phage system. To investigate the native phage system, genomic DNA of *Synechococcus* phage S-B43 was purified and digested to nucleosides. LC–MS analysis confirmed the presence of 5gcxmdC (Fig. 7), but not 5cxmdC (Fig. 7a), demonstrating the relevance of the

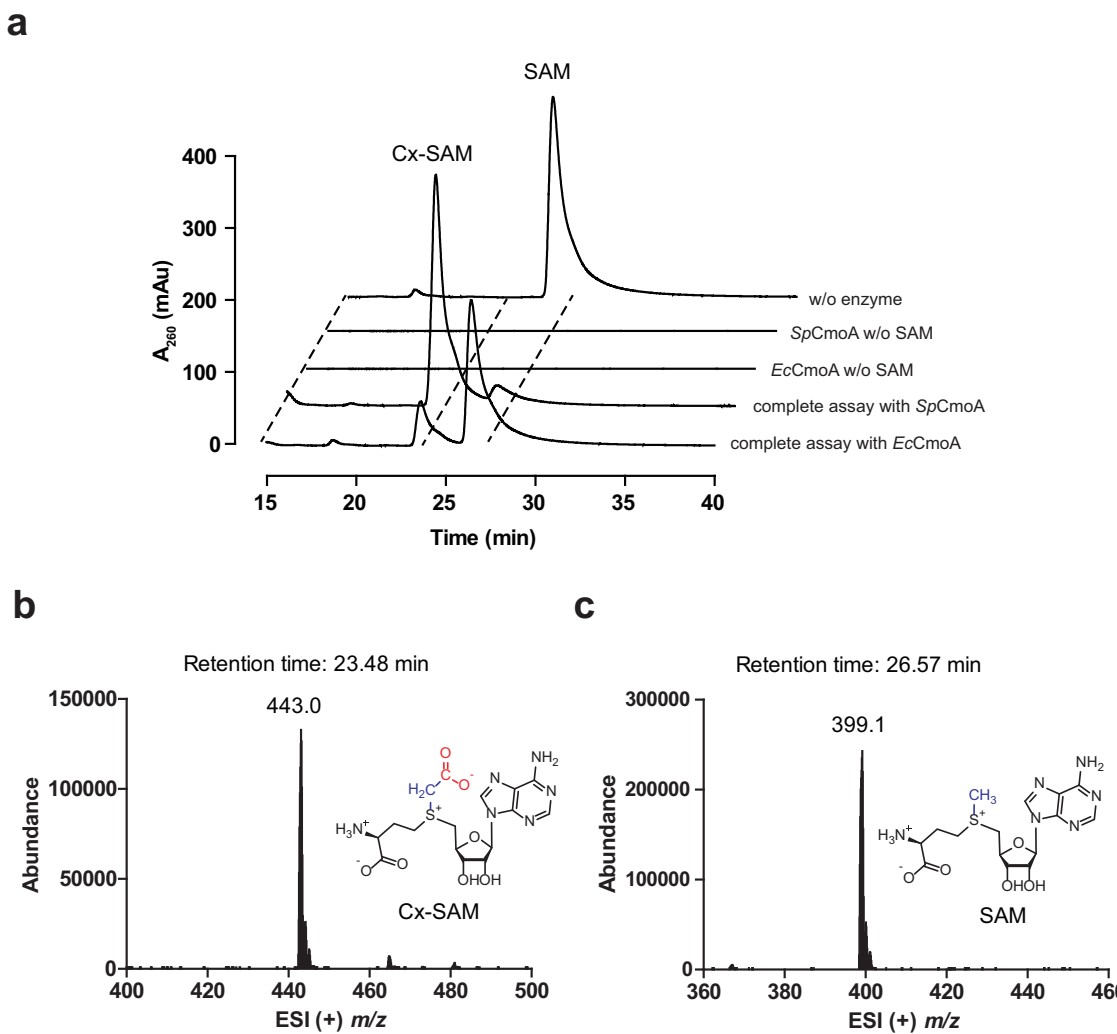

**Fig. 2 | The activity assay of SpCmoA. a** HPLC elution profiles of assays of SpCmoA and EcCmoA, monitoring the absorbance at 260 nm. ESI (+) $m/z$ spectra corresponding to EIC peaks for (**b**) Cx-SAM ($m/z = 443.0$) and (**c**) SAM ($m/z = 399.1$).

5gcxmC modification in the native phage. An estimation based on the integrative peaks suggested that 10% of C bases and 75% of the GC sequences in the phage genome are modified.

### Bioinformatics investigation of CmoX and CmoY homologs in phages

The identification of Cx-SAM-binding residues in the crystal structure enabled the search for candidate DNA cytosine carboxymethyl-transferases in phage-derived genome sequences. Using SpCmoX as a query, homologous sequences were retrieved from IMGVR v4, Uni-ProtKB, and a custom database prepared by extracting phage sequences from latest metagenomic data in MGnify, using jackhmmer (E-value $10^{-5}$, 3 iterations). This search yielded 5,615 candidate CmoX sequences containing both the 'SPPC' motif (S95 P96 P97 C98) and the conserved residue R350, which is involved in carboxymethyl group binding. A Sequence Similarity Network (SSN) was constructed for these CmoX sequences to visualize their sequence diversity and genomic context (Fig. 8a–c). Multiple sequence alignments and weblogo analyses revealed conservation of the active site E and R residues, which are positioned to interact with the $N^4$ and $O^2$ atoms of cytosine, respectively (Fig. 8d, e).

Many contigs contain a ParBc domain protein (PF02195), which is a small DNA-binding domain that frequently occurs as a fusion to various DNA methyltransferases[23], although its precise function in this context remains unknown (Fig. 8c). CmoA and AMP-forming "Ligase" genes were identified within these contigs using both single-protein searches and Pfam family searches, and the results were merged. In total, 4257 CmoA, 2251 ligase, and 1660 ParBc sequences were identified (Fig. 8a–c). A SSN was constructed for the AMP-forming ligases to visualize their sequence diversity (Fig. 8f). The largest cluster contains A0A514ABJ0 (SpCmoY), and structural modeling indicates that members of this cluster, along with several smaller clusters, contain active site residues involved in binding the glycine carboxylate group (Supplementary Fig. 8a, b), suggesting that the substrate may be glycine or another α-amino acid. The second largest cluster contains A0A7T0Q3R0, and these proteins possess a glutamine-hydrolyzing amidase domain, homologous to that of asparagine synthetase, suggesting that their substrate is glutamine-derived ammonia. Unfortunately, attempts to obtain soluble recombinant enzyme from this cluster were unsuccessful. Several other clusters contain distinct active site residues, indicating the possibility of alternative amine substrates that remain to be identified (Supplementary Fig. 8c).

### Discussion

Cx-SAM was first identified as a coenzyme involved in RNA modification[15] and later shown to be utilized by bacteria in natural product biosynthesis[24]. Our study of Synechococcus phage S-B43 shows that Cx-SAM also plays a role in the distinct DNA modification 5cxmC,

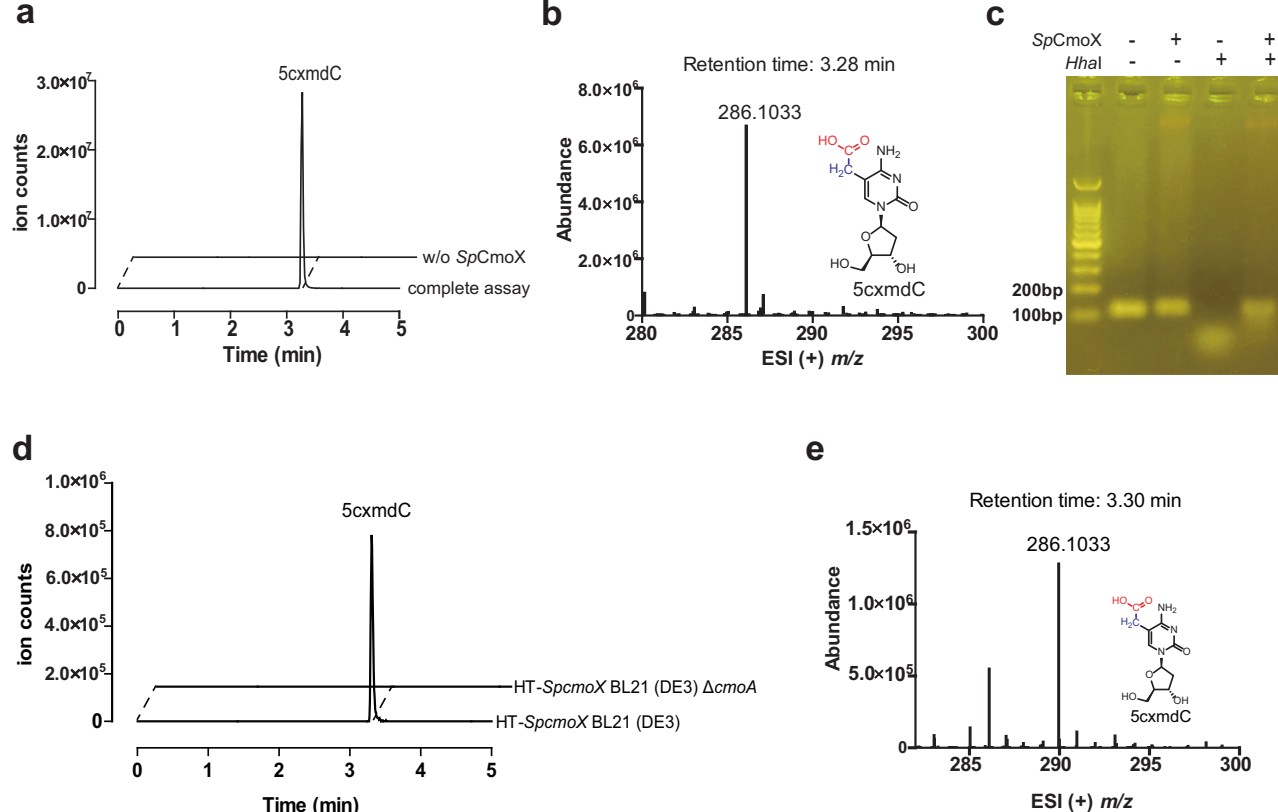

**Fig. 3 | The activity assay and physiological function of SpCmoX in vitro and in vivo. a** LC–MS extracted ion chromatogram of the digested dsDNA product, monitoring the formation of 5cxmdC ($m/z = 286.1$). **b** ESI (+) $m/z$ spectrum corresponding to the EIC peak for 5cxmdC. **c** Agarose gel analysis of HhaI digestion of the SpCmoX-treated dsDNA. This experiment was independently repeated three times with consistent results. **d** LC–MS extracted ion chromatograms detecting the presence of 5cxmdC ($m/z = 286.1$) from digested HT-SpcmoX plasmids extracted from BL21 (DE3) and BL21 (DE3) ΔcmoA cells. **e** The ESI (+) $m/z$ spectrum corresponding to EIC peak for 5cxmdC.

installed by a phage-encoded cytosine C5-methyltransferase homolog, CmoX. Thus, just as bacteria have adopted carboxy-SAM for secondary metabolite synthesis in the context of interspecies competition[24], phages have recruited it for DNA modification in their evolutionary arms race with bacterial hosts, providing an example of how phages co-opt host RNA modification machinery to expand their DNA modification repertoire[10]. Notably, the phage-encoded CmoA exhibits higher activity than its bacterial counterpart, which may reflect the increased demand for Cx-SAM to achieve genome-wide modification. Further investigation is needed into the availability of prephenate, the co-substrate for CmoA and a key shikimate pathway intermediate[25], across different bacterial hosts. Additionally, since some phages encode SAM lyases that disrupt host SAM-dependent defenses such as BREX[26], it warrants investigation whether CmoA-mediated SAM depletion similarly impacts host metabolism and signaling, including the production of antiviral messengers like SAM-AMP[27].

Unlike typical DNA methyltransferases that install a methyl group as a steric marker, CmoX introduces a reactive carboxylate handle that enables subsequent hypermodification, in this case via ligation of glycine. The crystal structure of CmoX reveals an active-site arginine that interacts with the substrate carboxymethyl group, and strikingly mirrors a previously reported DNA methyltransferase variant M.MpeI N374K, which was engineered to accept Cx-SAM. While the engineered carboxymethyltransferase preferentially targets CG motifs[21], CmoX exhibits complementary sequence specificity, recognizing GC motifs. Insights from the crystal structure enabled the identification of additional candidate carboxymethyltransferases in phages, although further work is needed to determine their sequence specificities.

To date, the most chemically diverse DNA modification–hypermodification systems involve the installation of a hydroxyl group in 5-hydroxymethyl-dU or -dC, serving as a handle for subsequent transformations such as glycosylation, amination, and carbamoylation[28,29]. In contrast, the carboxymethyltransferase system described here installs a carboxyl group as a complementary and versatile handle for amide bond formation. Our bioinformatics analyses suggest the presence of additional amide ligases associated with CmoA and CmoX, and further investigation is needed to uncover the full spectrum of amide modifications in natural phages. Apart from protection against restriction endonucleases, several natural and synthetic DNA modifications have been shown to either enhance or inhibit transcription by RNA polymerases, and reactive derivatives of hydroxymethyl-dC and dU have been utilized to achieve enzymatic or photochemical control of transcription[30–34]. Further studies are warranted to determine how 5cxmdC and its derivatives affect RNA polymerase activity and to explore their potential applications in transcriptional regulation.

The strategy for DNA functionalization in phage S-B43 mirrors earlier engineering approaches using DNA, RNA, or protein methyltransferases and propargyl analogs of SAM, to enable subsequent modification via click chemistry[35–37]. Identification of variants of CmoX and ligase in diverse phages highlights the potential for applications of such systems in DNA modification. While CmoX mediates global DNA modification to evade restriction systems, enzyme engineering could produce carboxymethyltransferases with extended recognition sequences, enabling site-specific DNA modification. Although our preliminary search did not identify candidate

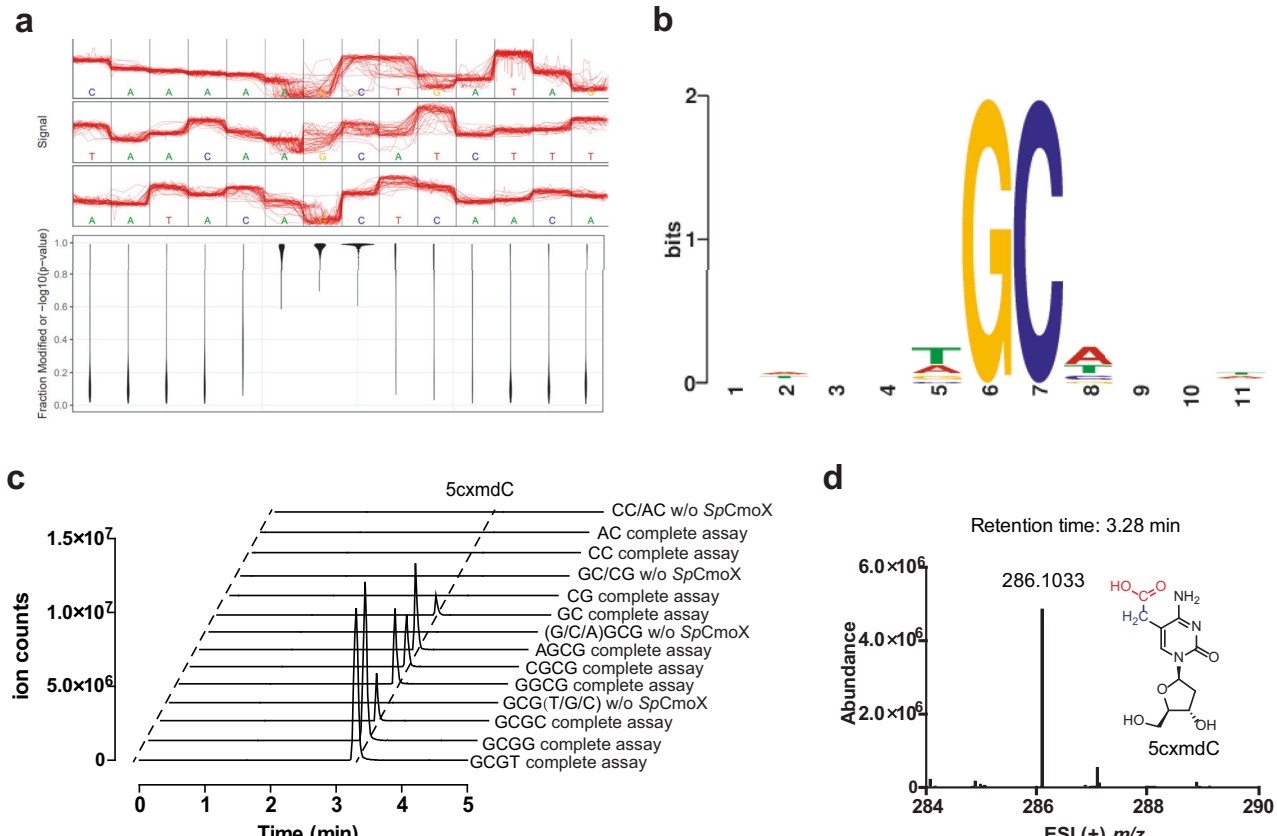

**Fig. 4 | Nanopore sequencing of *Synechococcus* phage S-B43 genome and sequence specificity for *Sp*CmoX. a** Plot of the estimation results of the relative position of the modified base in the enriched motif. **b** Motif predicted by MEME from significantly modified regions of the *Synechococcus* phage S-B43 genome. To facilitate motif discovery, five positions on either side of the most modified base per region were included. **c** LC–MS extracted ion chromatograms monitoring the formation of 5cxmdC ($m/z$ = 286.1) in various assays. The substrate dsDNA was prepared by annealing 25 bp palindromic oligonucleotides (Supplementary Table 3). In the enzyme-free negative controls, GCG(T/G/C) and (G/C/A)GCG, mixed annealed oligos of the three preceding assays, were used as substrates. **d** ESI (+) $m/z$ spectrum corresponding to EIC peak for 5cxmdC.

adenine N6- or cytosine N4-carboxymethyltransferases associated with CmoA in phages, such activities could theoretically be realized through enzyme engineering. Finally, engineering the ligase to accept a broader range of amine substrates could enable the incorporation of diverse functional groups, such as bioorthogonal handles for click chemistry, for applications in sequencing, DNA therapeutics, and chemical biology.

## Methods
### Materials and general methods
Lysogeny broth (LB) medium was prepared with yeast extract and tryptone (Oxoid, England). Ultrapure deionized (DI) water used in this work was prepared using Millipore Direct-Q. Ultrapure methanol and acetonitrile, used for liquid chromatography–mass spectrometry (LC–MS), were purchased from Concord Technology Co., Ltd (China). Prephenate barium salt was purchased from Sigma-Aldrich, Merck (U.S.A.). *S*-adenosyl-L-methionine (SAM) was acquired from Shanghai Yuanye Bio-Technology Co., Ltd (China). Oligonucleotide primers were synthesized by GENEWIZ, Inc. (Beijing, China). All protein purification chromatographic experiments were performed on liquid chromatography (FPLC) systems, including the ÄKTA pure, ÄKTA prime plus (GE Healthcare, U.S.A.) and UEV-25L (L1) system (Union-Biotech, China), equipped with appropriate columns (GE Healthcare, U.S.A.). Protein concentrations were calculated from the absorption at 280 nm, measured using MiuLab Ultra Trace Ultraviolet Spectrophotometer ND-100C (Hangzhou, China) or Eppendorf Biophotometer D30 (Hamburg, Germany).

### Gene synthesis, molecular cloning and plasmid construction
*Escherichia coli* codon-optimized DNA fragments, encoding *Synechococcus* phage S-B43 CmoA (*Sp*CmoA, A0A514ABH6), DNA cytosine carboxymethyltransferase (*Sp*CmoX, A0A514ABG7) and ATP-dependent amide ligase (*Sp*CmoY, A0A514ABJ0), were synthesized and inserted into the *Ssp*I restriction site of the modified pET28a (+)-HT vector by General Biosystems, Inc (Anhui, China). The resulting plasmids (HT-*Spcmo*A, HT-*Spcmo*X, HT-*Spcmo*Y) contain an N-terminal His$_6$-tag and a Tobacco Etch Virus (TEV) protease cleavage site, followed by the gene of interest. *Spcmo*A was also amplified from the HT-*Spcmo*A using p1F and p1R and inserted into the modified pET28a (+)-HMT vector at the *Ssp*I restriction site via Gibson assembly. The resulting plasmid (HMT-*Spcmo*A) contains in tandem: an N-terminus His$_6$-tag, maltose binding protein (MBP) and a TEV protease cleavage site, followed by *Spcmo*A. The gene for *E. coli* (strain K12) CmoA (*Ec*CmoA, P76290) was amplified by colony PCR with primers p2F and p2R and inserted into the pET28a (+)-HT vector (HT-*Eccmo*A) for expression with an N-terminal His$_6$-tag via Gibson assembly. The plasmid HT-*Spcmo*X-*Spcmo*Y was constructed by amplifying the HT-*Spcmo*X vector with primers p3F and p3R and inserting the *Spcmo*Y gene amplified from HT-*Spcmo*Y using primers p4F and p4R via Gibson assembly (Supplementary Table 2). All plasmids were confirmed by DNA sequencing.

### Protein expression, purification and quantification
HMT-*Spcmo*A, HT-*Spcmo*X, HT-*Spcmo*Y and HT-*Eccmo*A plasmids were transformed into BL21 (DE3) *E. coli* cells. Transformants were selected

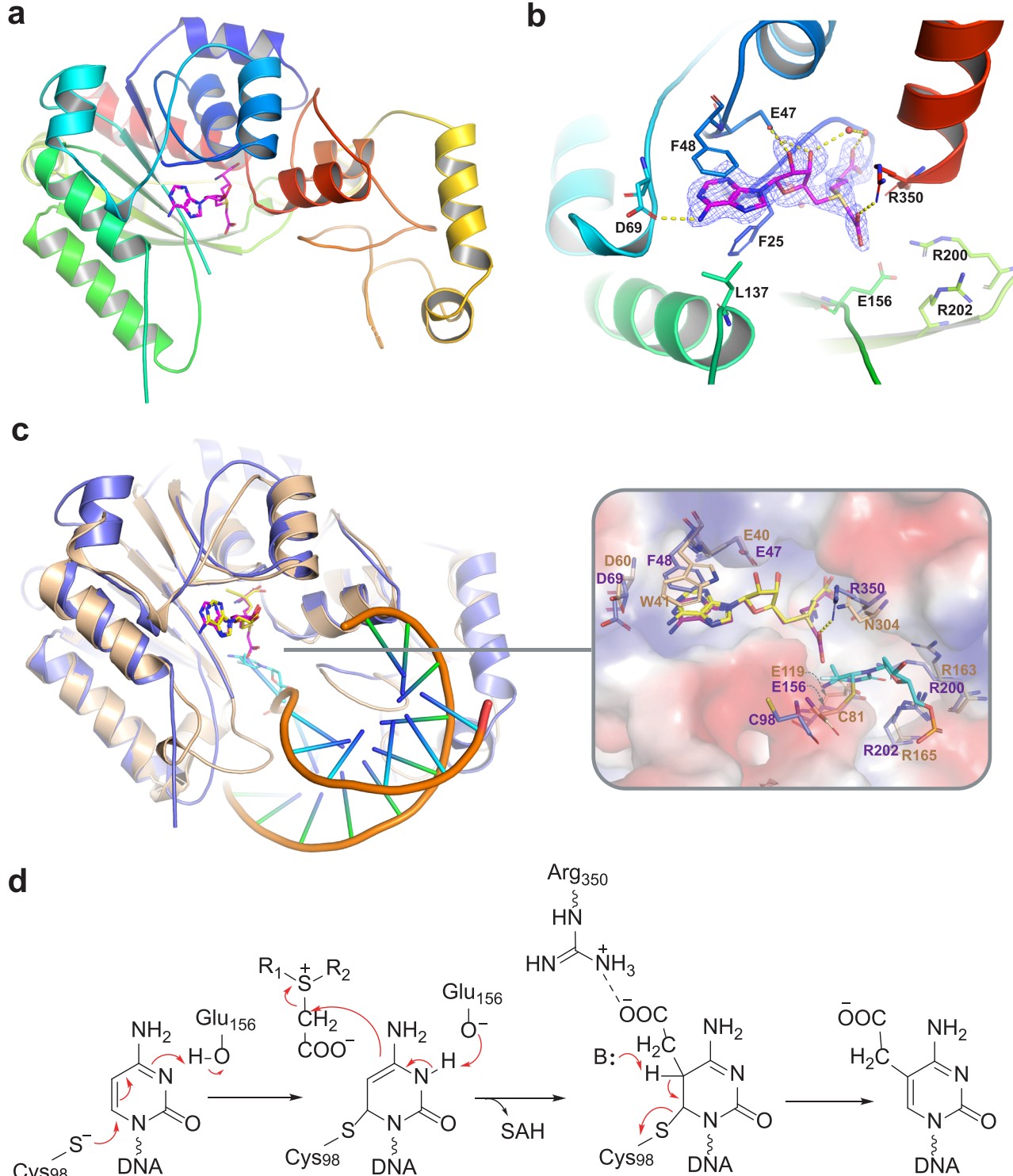

**Fig. 5 | X-ray crystal structure of SpCmoX in complex with Cx-SAM. a** Overall structure of the *Sp*CmoX monomer. **b** Interactions between Cx-SAM and active site residues of *Sp*CmoX. Hydrogen bonds are represented by dashed lines. The $F_o$-$F_c$ omit map is shown in black mesh, contoured at 3.0σ around the Cx-SAM binding site. **c** Overlay of the overall structures and active sites of *Sp*CmoX (slate blue, PDB entry 8YF3) and HhaIM (wheat, PDB entry 1MHT). Cα-based superposition of the two structures shows an RMSD of 1.027 Å. Key residues involved in substrate and cofactor binding are shown. **d** Proposed mechanism for the formation of 5cxmdC in the active site of *Sp*CmoX.

on LB agar plates with 50 μg/mL kanamycin. After an overnight incubation at 37 °C, a single colony was inoculated into 5 mL of LB medium, supplemented with 50 μg/mL kanamycin, as starter culture. Cells were grown at 37 °C for 5 h, before being transferred into 1 L of the same medium at 37 °C with constant shaking at 220 rpm until $OD_{600}$ reached 0.6–0.8. The temperature was then decreased to 18 °C, followed by the

addition of isopropyl β-D-1-thiogalactopyranoside (IPTG) to a final concentration of 0.3 mM to induce protein expression. After continuous shaking for 16 h, the cells were harvested by centrifugation (5,000 × *g*, 10 min, 4 °C). The resulting cell pellets for *Sp*CmoA, *Sp*CmoY and *Ec*CmoA were then resuspended in lysis buffer containing Tris-HCl (50 mM, pH 8.0), KCl (150 mM), phenylmethanesulfonyl

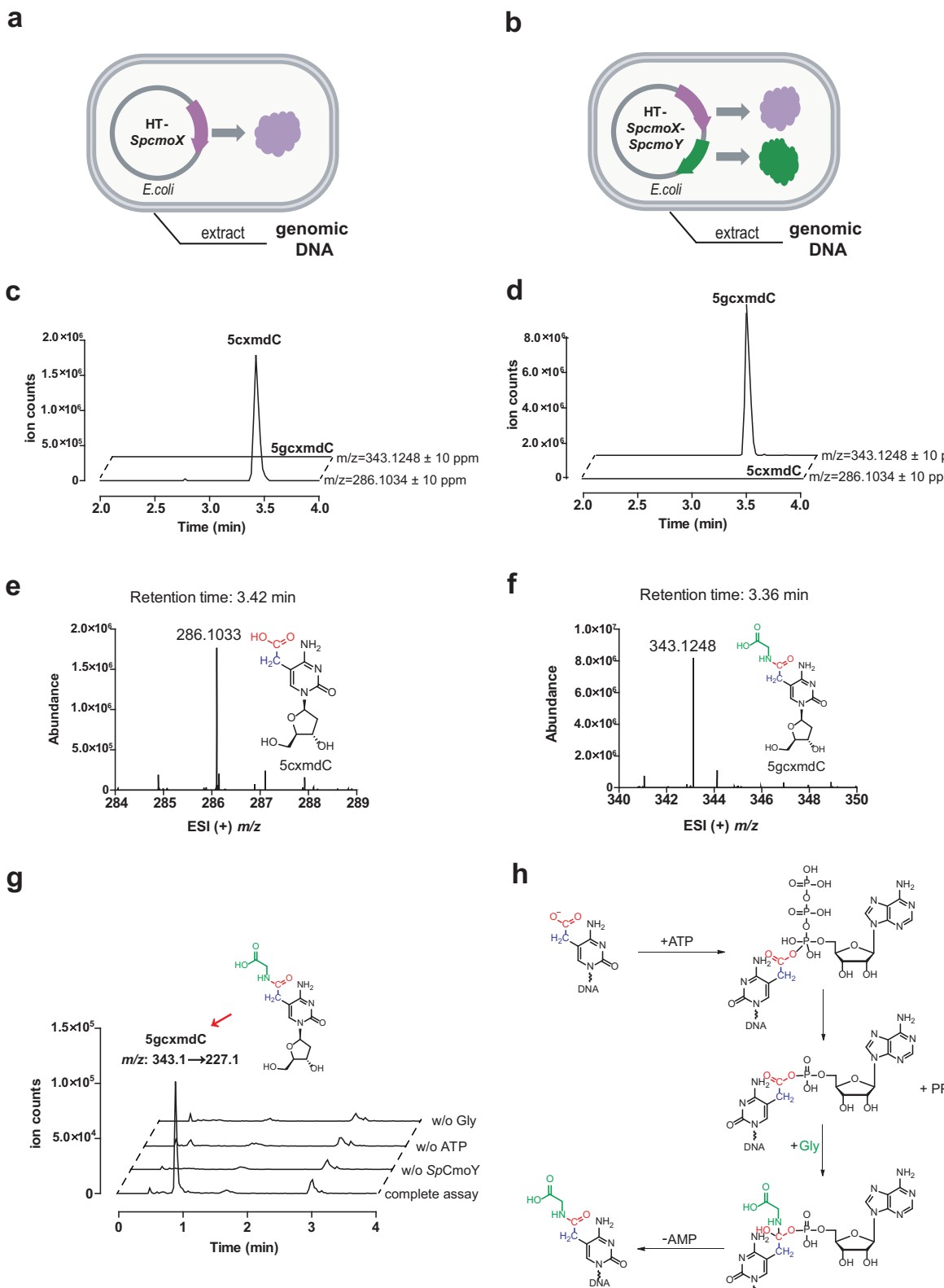

**Fig. 6 | The activity assay of *Sp*CmoY. a** Extraction of the genomic DNA from *E. coli* cells overexpressing *Sp*CmoX. **b** Extraction of the genomic DNA from *E. coli* cells overexpressing both *Sp*CmoX and *Sp*CmoY. **c** LC–MS extracted ion chromatograms monitoring the formation of 5cxmdC ($m/z = 286.1$) and 5gcxmdC ($m/z = 343.1$) in the genomic DNA of (**a**). **d** LC–MS extracted ion chromatograms monitoring the formation of 5cxmdC ($m/z = 286.1$) and 5gcxmdC ($m/z = 343.1$) in the genomic DNA of (**b**). **e** ESI (+) $m/z$ spectrum corresponding to EIC peak for 5cxmdC in (**c**). **f** ESI (+) $m/z$ spectrum corresponding to EIC peak for 5gcxmdC in (**d**). **g** In vitro *Sp*CmoY enzyme activity assay with the genomic DNA of (**a**) as the substrate. LC–MS/MS extracted ion chromatograms were used to monitor the formation of 5gcxmdC ($m/z = 343.1 \rightarrow 227.1$). **h** Proposed model for the *Sp*CmoY-catalyzed reaction.

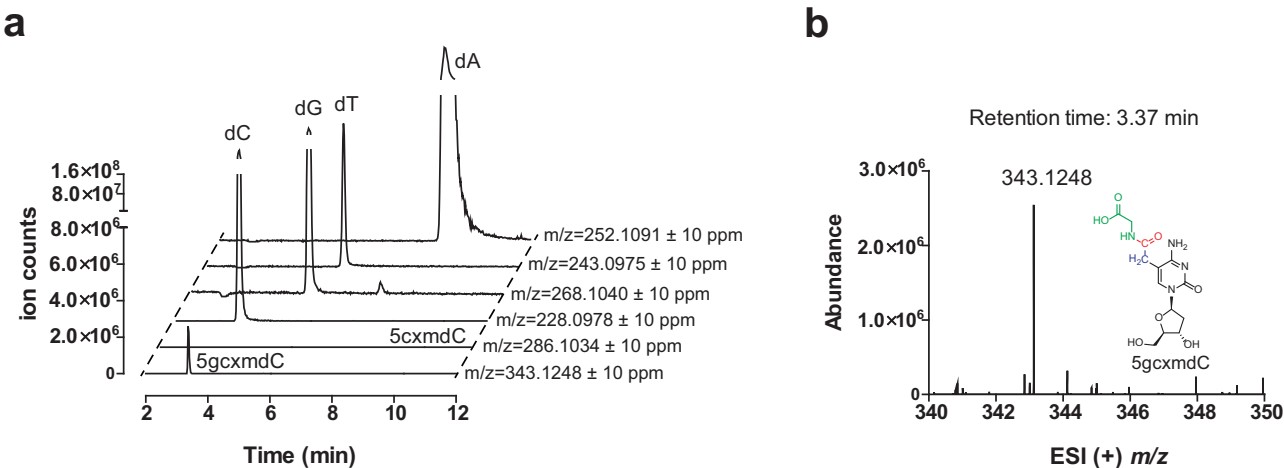

**Fig. 7 | Validation of 5gcxmC incorporation in the genome of *Synechococcus* Phage S-B43. a** LC–MS extracted ion chromatogram of the digested *Synechococcus* Phage S-B43 genome, monitoring the formation of 5gcxmdC (*m/z* = 343.1). **b** ESI (+) *m/z* spectrum corresponding to the EIC peak for 5gcxmdC.

fluoride (PMSF, 1 mM), lysozyme (0.2 mg/mL), 0.03% Triton X-100, DNase I (0.02 mg/mL) and frozen in a −80 °C freezer. The cell pellet for *Sp*CmoX was resuspended in a different lysis buffer containing HEPES (20 mM, pH 6.8), NaCl (250 mM), PMSF (1 mM), lysozyme (0.2 mg/mL), 0.03% Triton X-100 and DNase I (0.02 mg/mL) before frozen. The cell suspensions were thawed and incubated at room temperature (RT) for 50 min to allow lysis. Nucleic acids were precipitated by addition of 6% streptomycin sulfate stock solution (final concentration, 1%), followed by centrifugation (12,000 × *g*, 10 min, 4 °C) to remove precipitated nucleic acids and cell debris. For all protein samples except *Sp*CmoX, the supernatant was filtered and loaded onto a 5 mL TALON Co²⁺ column (Takara Bio USA, Inc.), pre-equilibrated with a solution of Tris-HCl (20 mM, pH 7.5) and KCl (200 mM), containing β-mercaptoethanol (BME, 5 mM) (Buffer A). The column was then washed with 10 column volumes (CV) of Buffer A and the protein was eluted with 5 CV of Buffer A containing 150 mM imidazole. The eluted protein (30 mL) was dialyzed against 2 L of Buffer A at 4 °C for 4 h to remove the imidazole. The purification of *Sp*CmoX employed Buffer B, containing HEPES (20 mM, pH 6.8) and 250 mM NaCl. The purified proteins were examined on an 8–16% SDS polyacrylamide gradient gel, with a commercial protein marker (Genstar, Beijing) and visualized by Coomassie staining. The protein samples were then aliquoted and glycerol was added to a final concentration of 10%. Lastly, the samples were flash frozen in liquid N₂ before being stored at −80 °C. Additional chromatographic steps were performed to further purify *Sp*CmoX for protein crystallization. The eluate from the TALON Co²⁺ column was diluted with Buffer C (20 mM HEPES, pH 6.8) to a NaCl concentration of 100 mM before loaded onto a 5 mL Heparin Sepharose high performance column. The column was eluted with 60 CV linear salt gradient from 100–675 mM NaCl in Buffer C. *Sp*CmoX was collected, concentrated and buffer-exchanged with Buffer B containing 1 mM tris (2-carboxyethyl) phosphine (TCEP). The concentrations of the proteins were calculated based on their absorptions at 280 nm, using extinction coefficient values [*Sp*CmoA ($\varepsilon_{280}$ = 106,230 M⁻¹ cm⁻¹), *Sp*CmoX ($\varepsilon_{280}$ = 34,840 M⁻¹ cm⁻¹), *Sp*CmoY ($\varepsilon_{280}$ = 31,860 M⁻¹ cm⁻¹), *Ec*CmoA $\varepsilon_{280}$ = 19,940 M⁻¹ cm⁻¹)].

### Analysis of the oligomeric state of *Sp*CmoX

The oligomeric state of *Sp*CmoX was investigated using size exclusion chromatography (SEC). Briefly, a solution of *Sp*CmoX (5 mL, 6.5 mg/mL) was injected into a Superdex 200 gel filtration column (300 mL) and eluted over 170 min with Buffer B containing 1 mM dithiothreitol (DTT) at 2 mL/min. The same conditions were used to analyze a mixture of molecular weight markers (Sigma MWGF 1000-1KT) to establish a standard curve using a second-degree polynomial for the relationship between log(molecular weight) and retention volume. The observed molecular weight for *Sp*CmoX was 42.9 kDa indicating that it exists as a monomer (43.1 kDa) in solution.

### LC–MS analysis of *Sp*CmoA activity

A 200 μL reaction mixture containing 10 μM of *Sp*CmoA, 5 mM SAM and 5 mM prephenate in 20 mM sodium phosphate buffer, pH 6.8 was incubated at 37 °C for 1 h. This assay was performed with *Ec*CmoA as a positive control, and without SAM or *Sp*CmoA as negative controls. At the end of the reaction, an equal volume of acetonitrile was added to precipitate the protein. After removing the precipitate using a 0.22 μm nylon membrane filter, the filtrate was analyzed using LC–MS (Agilent 6420 Triple Quadrupole LC–MS instrument; ESI positive ion mode detection; Merck ZIC-HILIC column, bead size of 5 mm, pore size of 200 Å, 150 × 4.6 mm) with a 9:1 (v/v) 0.1 M ammonium acetate/acetonitrile and acetonitrile gradient solvent system (90–70% vol. acetonitrile over 15 min, 70–50% vol. acetonitrile over 25 min, and 50% acetonitrile over 5 min, sample injection volume: 20 μL, UV 260 nm detection, flow rate: 0.75 mL/min). To compare the activities of *Sp*CmoA and *Ec*CmoA, enzymatic reactions each in a volume of 400 μL containing 20 mM sodium phosphate buffer (pH 6.8), 2 μM CmoA, 10 mM SAM, and 10 mM prephenate were incubated at 37 °C. At 2-min intervals, 40 μL of the reaction mixture was sampled and immediately quenched with an equal volume of acetonitrile. The samples were then centrifuged to remove the enzyme before injection into the LC–MS system for peak integrative analysis referring to a standard curve established with known concentrations of chemically synthesized caboxy-SAM. Each assay was performed in triplicate.

### Chemical and enzymatic synthesis of Cx-SAM

Chemical synthesis of Cx-SAM was performed as previously described[38]. Briefly, 5.0 mg of SAH was dissolved in 0.8 mL of 150 mM ammonium bicarbonate (NH₄HCO₃). Subsequently, 166.7 mg of 2-iodoacetic acid was added. The reaction mixture was incubated at 37 °C for 24 h with constant agitation. Following incubation, 20 mL of methanol was added, and the solution was maintained at 4 °C overnight. The resulting precipitate was collected by centrifugation at 2000 × *g* for 30 min at 4 °C, followed by two washes with cold methanol and dissolved in nuclease-free water. Cx-SAM was purified using an LC–MS system (Agilent 6420 Triple Quadrupole LC–MS) equipped with an electrospray ionization (ESI) source in positive ion mode. Separation was performed on a Merck ZIC-HILIC column (5 μm

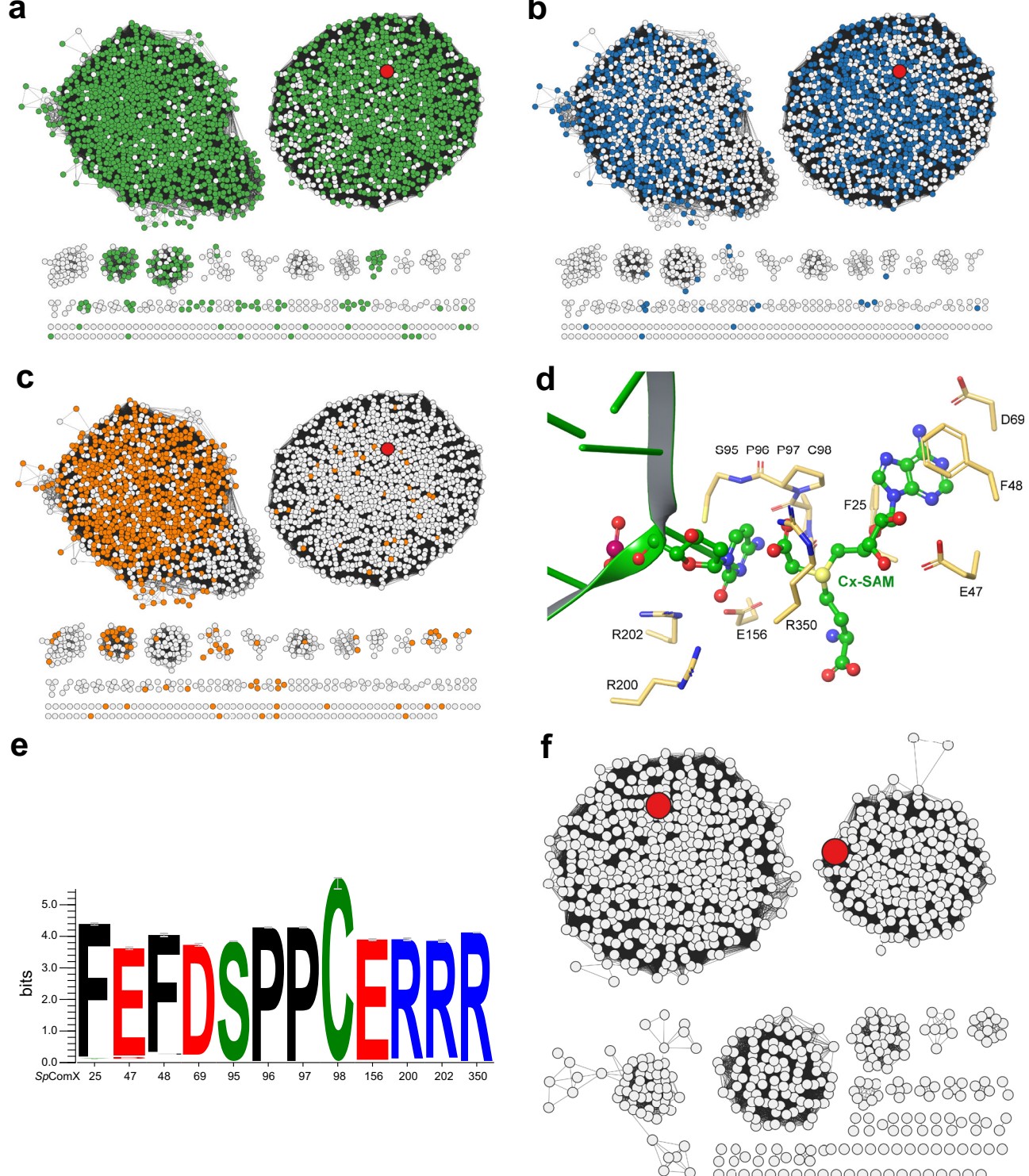

**Fig. 8 | Bioinformatic support for a two-step cytosine modification pathway in phages. a–c** The SSN of 5,615 CmoXs displayed at the *e*-value threshold of 10⁻⁷⁰ (~50% sequence identity). The node coloring indicates genomic context gene associations: CmoA, AMP-forming ligase and ParBc-domain protein (PF02195), respectively. The large red node in each SSN represents *Sp*CmoX. **d** Predicted structure model of *Sp*CmoX bound to Cx-SAM, showing the ligand binding pocket.

The structure was generated using AlphaFold 3 and used to visualize the predicted binding mode of Cx-SAM. **e** Weblogo of active site residues of those CmoXs containing the SPPC motif and the conserved R350 residue. **f** The SSN of 2,251 AMP-forming ligases displayed at the *e*-value threshold of 10⁻⁵⁶ (~30% sequence identity). A0A514ABJ0 (*Sp*CmoY) and A0A7T0Q3R0 are represented by large red nodes.

bead size, 200 Å pore size, 150 × 4.6 mm) using a gradient solvent system of 0.1 M ammonium acetate/acetonitrile (9:1, v/v) and acetonitrile, as previously described. Cx-SAM concentration was determined by absorbance at 260 nm ($\varepsilon = 15{,}400\ \text{L·mol}^{-1}\text{·cm}^{-1}$). We note that the chemical synthesis yields a mixture of stereoisomers at the

sulfur center[39]. For non-quantitative assays of *Sp*CmoX activity, Cx-SAM was prepared by a *Sp*CmoA-catalyzed reaction. A reaction mixture (200 µL) containing HEPES buffer (20 mM, pH 6.8), NaCl (250 mM), *Sp*CmoA (10 µM), SAM (10 mM) and prephenate (10 mM) was incubated at 37 °C for 1 h.

## LC−MS analysis of SpCmoX activity

A reaction mixture (200 μL) containing HEPES buffer (20 mM, pH 6.8), NaCl (250 mM), DTT (1 mM), Cx-SAM reaction mixture (40 μL), SpCmoX (10 μM) and double-stranded DNA (dsDNA) fragment (10 μg, 5′-TGGCGAATGGGACGCGCCCTGTAGCGGCGCATTAAGCGCGGCGGG TGTGGTGGTTACGCGCAGCGTGACCGCTACACTTGCCAGCGCCCTAG CGCCCGCTCCTTTCGCTT-3′), amplified from modified pET28a (+)-HT vector using p5F and p5R. After overnight incubation at 18 °C, the dsDNA fragment was purified from the crude reaction mixture using a DNA Fragment Purification Kit (Genstar, Beijing). 5 μg of the purified dsDNA fragment was then enzymatically digested with the Nucleoside Digestion Mix (New England Biolabs, M0649S) in a 100 μL reaction mixture. The products were subjected to LC−MS analysis (Thermo Fisher Q Exactive HF/UltiMate 3000 RSLCnano; ESI positive ion mode detection; Syncronis aQ C18 column, particle size of 3 μm, 150 × 4.6 mm) with a 10 mM pH 4.6 aqueous ammonium acetate and methanol gradient solvent system (20-32% vol. methanol over 12 min, sample injection volume: 18 μL, flow rate: 0.5 mL/min). The eluate was detected by absorbance at 260 nm and subjected to MS-MS analyses using the same machine and elution protocol.

## Restriction enzyme digestion of DNA with 5cxmC

The SpCmoX-treated dsDNA fragment was incubated with HhaI (New England Biolabs) and CutSmart buffer (New England Biolabs) at 37 °C according to the manufacturer's instruction. The reaction mixture was then separated on a 2% agarose gel with gel electrophoresis conditions at 150 V for 30 min. Controls for the same gel assay included samples omitting treatment of SpCmoX, HhaI or both.

## Assay of SpCmoX activity on annealed double-stranded palindromic oligo substrates

Palindromic oligos listed in Supplementary Table 3 were designed by us and synthesized by GENEWIZ. A 100 μL reaction mixture containing Tris-HCl (10 mM, pH 8.0), NaCl (50 mM), palindromic oligo (100 μM) was incubated at 95 °C for 5 min, before being cooled to RT. The annealed dsDNA was stored at −20 °C till use. A reaction mixture (100 μL) containing HEPES (20 mM, pH 6.8), NaCl (250 mM), Cx-SAM reaction mixture (10 μL), DTT (1 mM), SpCmoX (10 μM) and annealed double-stranded palindromic oligo (25 μL) was incubated at 18 °C overnight. The assay was performed without SpCmoX as negative control. At the end of the reaction, the mixture was heated at 95 °C for 5 min to precipitate the protein. The precipitate was removed by centrifugation and the supernatant was enzymatically digested with the Nucleoside Digestion Mix (New England Biolabs, M0649S) in a 100 μL reaction mixture. LC−MS analysis was performed as described earlier.

## Construction of the ΔcmoA E. coli strain

The BL21 (DE3) ΔcmoA E. coli strain was constructed using a CRISPR-aided homologous recombination strategy. Briefly, primer pairs p6F/p6R and p7F/p7R (Supplementary Table 2) were used to amplify the 500-bp region upstream and the 504-bp region downstream of EccmoA to produce fragments 1 and 2, respectively. Fragments 1 and 2 were then assembled using PCR with primers p6F and p7R to produce fragment 3. With the pRed_Cas9_recA plasmid as template, fragments 4 and 5 were produced by PCR amplification using primer pairs p8F/p8R and p9F/p9R (Supplementary Table 2). Fragments 3, 4 and 5 were assembled using Gibson assembly to produce the recombinant pRed_Cas9_recA_ΔcmoA plasmid and confirmed by sequencing, which contained a guide RNA sequence (5′-CGCCATTGAAAACGCATCGA-3′) to introduce a Cas9-catalyzed cleavage near the homologous recombination site, thereby increasing the rate of the desired homologous recombination[40]. pRed_Cas9_recA_ΔcmoA plasmid was transformed into BL21 (DE3) cells and plated on LB agar, supplemented with 50 μg/mL kanamycin. After overnight incubation at 30 °C, a colony was

inoculated into 5 mL of LB medium, supplemented with 50 μg/mL kanamycin. Cells were grown at 30 °C for 2 h before the addition of 2 g/L L-arabinose to induce Cas9 and gRNA expression. After 24 h, the cells were harvested and streaked onto LB agar, supplemented with 50 μg/mL kanamycin. The agar plate was then incubated at 37 °C, a non-permissive temperature resulting in the ejection of the plasmid. Surviving colonies were selected as candidates for EccmoA knockout via homologous recombination and were further verified by colony PCR using primers p10F and p10R and confirmed by sequencing (Supplementary Table 2).

## SpCmoX activity in a recombinant E. coli system

HT-SpcmoX was transformed into BL21 (DE3) WT and ΔcmoA cells, followed by induction of SpCmoX expression. The plasmids were extracted and purified using the TIANprep Mini Plasmid Kit (TIAN-GEN). 10 μg each was enzymatically digested with the Nucleoside Digestion Mix (New England Biolabs, M0649S) in a 100 μL reaction volume and subjected to the LC−MS analysis as described in the LC−MS analysis of SpCmoX activity section of the Methods.

## Crystallization, data collection and structure determination of SpCmoX

Initial screening of SpCmoX crystals was done by an automated liquid handling robotic system (Gryphon, Art Robbins), utilizing the sitting-drop vapor diffusion method in a 96-well format, at 291 K using various sparse matrix crystal screening kits (Hampton Research and Molecular Dimensions). The diffraction quality crystals co-crystallized with Cx-SAM were obtained by mixing SpCmoX (8.0 mg/mL) with BICINE (100 mM, pH 9.0), 20% (w/v) PEG 6000 and 1 mM Cx-SAM. The crystals were cryoprotected by the addition of 25% glycerol before being flash cooled in liquid nitrogen. Diffraction data were collected on a BL10U1 at the Shanghai Synchrotron Radiation Facility (SSRF) and processed to 1.9 Å resolution using HKL3000. Molecular replacement was performed by PHENIX(Version 1.14-3247)-Phaser[41], using a homology model of SpCmoX created by AlphaFold. The structure was manually built according to the modified experimental electron density using Coot (Version 0.8.9.2)[42] and further refined by PHENIX-Refine in iterative cycles. Supplementary Table 4 shows the statistics for data collection and final refinement. Protein structures were visualized and all structural figures were generated using PyMol (Version 3.1).

## Site-directed mutagenesis of SpCmoX and activity assays of the mutant enzyme

A point mutation in the active site of SpCmoX (R350N) was introduced by PCR using primers p11F and p11R with HT-SpcmoX as the template and confirmed by sequencing. The expression, purification, and activity assays of the mutant enzyme were performed as described for the wild-type enzyme.

## SpCmoY activity in a recombinant E. coli system

HT-SpcmoX-SpcmoY was transformed into BL21 (DE3) cells. The transformant was induced to express both proteins. Subsequently, genomic DNA was extracted and purified using the Bacteria Genomic DNA Kit from Beijing Zoman Biotechnology Co., Ltd (China) for analysis. The HT-SpcmoX transformant was used as a control. 10 μg each was enzymatically digested with the Nucleoside Digestion Mix (New England Biolabs, M0649S) in a 50 μL reaction volume and subjected to the LC−MS analysis as described in the LC−MS analysis of SpCmoX activity section of the Methods.

## LC−MS analysis of SpCmoY activity

Genomic DNA containing 5cxmC was extracted from the HT-SpcmoX transformant cells and used as the substrate for SpCmoY enzyme activity assays. A reaction mixture (200 μL) was prepared, containing HEPES (20 mM, pH 7.5), NaCl (250 mM), MgCl₂ (5 mM), ATP (5 mM),

Glycine (5 mM), 10 μM *Sp*CmoY, and 15 μg of genomic DNA. The reaction was incubated overnight at 18 °C. Following incubation, the genomic DNA was purified from the crude reaction mixture using the Bacteria Genomic DNA Kit (Beijing Zoman Biotechnology Co., Ltd, China). 5 μg of the purified genomic DNA was then enzymatically digested with the Nucleoside Digestion Mix (New England Biolabs, M0649S) in a 50 μL reaction mixture and subjected to quantitative mass spectrometry (LC−MS/MS). The nucleosides were separated by an SB-Aq C18 column (2.1 × 100 mm, 1.8 μm; Agilent Technologies) and detected by ABSCIEX QTRAP 6500 systems in positive ion mode. A gradient solvent system consisting of 5 mM aqueous ammonium acetate (pH 5.3) and acetonitrile was employed, with the acetonitrile concentration increasing from 0 to 50% over 7 min. The sample injection volume was 10 μL, and the flow rate was maintained at 0.4 mL/min. Negative controls included assays in the absence of *Sp*CmoY or one of the substrates, ATP or Glycine. The *Sp*CmoX-treated, 5cxmC-modified, 110-bp dsDNA fragment obtained by PCR amplification from pET28a (+)-HT vector was used as another substrate in the *Sp*CmoY assay.

### *Synechococcus* phage S-B43 genomic DNA extraction and analysis

*Synechococcus* sp. Strain MW02 (NCBI accession number KP113680) was cultivated in artificial seawater-based f/2 medium at 25 °C, under 25 μmol quanta m$^{-2}$s$^{-1}$ and a 12-h light: 12-h dark cycle. Phage expansion was performed by adding a solution of *Synechococcus* phage S-B43 (GenBank: MN018232.1) at a ratio of 1:9 (v/v) to the exponentially growing host *Synechococcus*. The phage-host suspension was then incubated at 25 °C, under constant irradiance of 25 μmol quanta m$^{-2}$s$^{-1}$ for 7 days until host cell lysis was observed based on the color and turbidity of the lysate[20]. The resulting lysate (200 mL) was centrifuged (7000 × *g*, 10 min, 4 °C) to remove most of the cell debris. 8.3% (w/v) PEG 6000 and 2% (w/v) NaCl were then added to the supernatant followed by incubation at 4 °C in the dark overnight. The phage and cell debris was then pelleted by centrifugation (15,800 × *g*, 15 min, 4 °C), and resuspended in 20 mL of SM buffer, containing Tris-HCl (50 mM, pH 7.5), NaCl (100 mM), magnesium sulfate (8.0 mM) and gelatin (0.1 g/L). The cell debris was then removed by differential centrifugation (10,000 × *g*, 10 min, 4 °C). The genomic DNA of *Synechococcus* phage S-B43 was extracted using a Lambda phage genomic DNA Kit from Beijing Zoman Biotechnology Co., Ltd (China). 5 μg genomic DNA was digested using the Nucleoside Digestion Mix (New England Biolabs, M0649S) in a 50 μL reaction volume, followed by LC−MS analysis as described in the LC−MS analysis of *Sp*CmoX activity section of the Methods.

### Nanopore sequencing and data analysis

3−4 μg of phage DNA per sample was used as input for library preparation. After the sample was qualified, size selection of long DNA fragments was performed using the PippinHT system (Sage Science, U.S.A.). The ends of the DNA fragments were then repaired followed by A-ligation reaction with the NEBNext Ultra II End Repair/dA-Tailing Kit (New England Biolabs, E7546). The adapter in the SQK-LSK109 (Oxford Nanopore Technologies, United Kingdom) was used for further ligation and the DNA library was measured by Qubit 4.0 Fluorometer (Invitrogen, U.S.A.). Approximately 700 ng of DNA library was constructed and performed on a Nanopore PromethION sequencer instrument (Oxford Nanopore Technologies, United Kingdom) at the Genome Center of Grandomics (Wuhan, China). The nanopore sequencing data was analyzed with the Tombo 1.5.1 software, using the protocol designed for de novo identification of DNA modification[43]. Modification sites, with significant values of at least 0.9, were chosen for the motif enrich analysis using the MEME suite 5.4.1[44]. The relative position of the modified base, with the highest significant value, was then estimated using Tombo 1.5.1.

### Bioinformatics

Sequence similarity network (SSN) analysis was carried out using the web-based Enzyme Function Initiative Enzyme Similarity Tool (EFI-EST)[45], and the results were visualized using Cytoscape version 3.8[46].

### Statistics and reproducibility

All experiments in this study except Nanopore sequencing and X-ray crystallography were repeated independently at least three times. For LC-MS traces, and agarose gel assays, representative results are shown.

### Reporting summary

Further information on research design is available in the Nature Portfolio Reporting Summary linked to this article.

## Data availability

The atomic coordinates and structure factors for the crystal structure of *Sp*CmoX in complex with Cx-SAM generated in this study have been deposited in the Protein Data Bank under accession code 8YF3. The large raw LC−MS and Nanopore FAST5 datasets can be shared by the corresponding author upon request. Other data supporting the findings of this study, including enzyme activity measurements, kinetic plots, LC−MS quantifications, and Nanopore modification analyses, are provided in the Source Data file. Source data are provided with this paper.

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

## Acknowledgements

This work was supported by the National Natural Science Foundation of China (NSFC) Distinguished Young Scholar of China Program (32125002, Y.Z.), the New Cornerstone Science Foundation (NCI2002321, Y.Z.), and the funding support from Agency for Science, Technology and Research (C211917011, Y.W.). We thank the instrument analytical center of School of Pharmaceutical Science and Technology at Tianjin University, in particular, Zhi Li and Yan Gao for providing assistance in our LC–MS analysis. We also thank the supports from the Core Facility of Basic Medical Sciences at Shanghai Jiao Tong University School of Medicine for LC–MS/MS data collection and analysis. We are grateful to Prof. Xiang Yu and Dr. You Wu, from the School of Life Sciences and Biotechnology at Shanghai Jiao Tong University, for their helpful discussions and analytical insights regarding Nanopore sequencing data. In addition, we thank Chuxiao Wang and Zhenquan Deng, Master students at the College of Marine Life Sciences, Ocean University of China, for their valuable assistance in phage cultivation and related experimental procedures, and doctoral student Hao Yu for his support in bioinformatics analyses.

## Author contributions

Q.Y., Lin.Z. and Y.L. conducted most of biochemical and molecular biology experiments. H.M. carried out the bioinformatic analyses. L.S. performed the Nanopore sequencing data analysis. L.L., Y.H., K.M., Y.C., Y.T., C.Z., and Y.W. participated in data acquisition. Q.Y., J.T., H.M., S.Z., M.W., Liang.Z., Y.W., Y. Zhang designed the experiments and wrote the paper.

## Competing interests

The authors declare no competing interests.

## Additional information

[1]New Cornerstone Science Laboratory, School of Pharmaceutical Science and Technology, Tianjin University, Tianjin, China. [2]Frontiers Science Center for Synthetic Biology (Ministry of Education), Tianjin University, Tianjin, China. [3]Key Laboratory of Systems Bioengineering (Ministry of Education), School of Chemical Engineering and Technology, Tianjin University, Tianjin, China. [4]School of Chemistry and Chemical Engineering, and Zhangjiang Institute for Advanced Study, Shanghai Jiao Tong University, Shanghai, China. [5]College of Marine Life Sciences, Ocean University of China, Qingdao, China. [6]iHuman Institute and School of Life Science and Technology, ShanghaiTech University, Shanghai, China. [7]Tianjin Institute of Industrial Biotechnology, Chinese Academy of Sciences, Tianjin, China. [8]Singapore Institute of Food and Biotechnology Innovation (SIFBI), Agency for Science, Technology and Research (A*STAR), 31 Biopolis Way, Nanos, Singapore, Singapore. [9]State Key Laboratory of Microbial Metabolism, School of Life Sciences and Biotechnology, Shanghai Jiao Tong University, Shanghai, China. [10]These authors contributed equally: Qiaoyu Yang, Lin Zhang, Yantao Liang.
✉e-mail: zhaosw@shanghaitech.edu.cn; mingwang@ouc.edu.cn; liangzhang2014@sjtu.edu.cn; weiyf@a-star.edu.sg; yan.zhang@tju.edu.cn

