## [Transparent Peer Review file · Nature Communications]

Carboxymethylcytosine is a natural base modification and a handle for bacteriophage DNA hypermodification

Corresponding Author: Professor Yan Zhang

Version 0:

Reviewer comments:

Reviewer #1

(Remarks to the Author)

This is a highly interesting, though very specialized, paper on the discovery of the biochemical pathway how bacteriophage S-B43 modifies its genome by carboxymethylation of C. The suitability to Nat. Comm. is a borderline case, because it is very very specialized field (the findings concern only one single bacteriophage), but the mechanism is so interesting that I still recommend publication in Nat. Comm.

The authors have identified the carboxymethyltransferase that transfers the carboxymethyl group from carboxy-SAM to DNA - this is a novel finding of this work, although the presence of the 5cxmC in the bacteriophage genome was known before. They succeeded in determination of X-ray structure of the enzyme with carboxy-SAM which sheds light on the mechanism and specificity. Not surprisingly, the 5cxmC modification protects the DNA from cleavage by RE. Very intriguing and interesting is the discussion of the repurposing of the bacterial enzymes for modification of the virus genome.

The experiments are performed well and the discussion is relevant. The only missing and potentially very interesting results would be the functional assay how the presence of the modification influences transcription - see below. Otherwise I like the work and enthusiastically recommend acceptance subject to revision:

Major point

Several previous works showed that some bacteriophage DNA modifications may not only serve as protection against RE cleavage but also as modulators or transcription. For examples, 5hmU and 5hm as well as some non-natural homologues (i.e. 5-ethylU) were found to enhance transcription by bacterial RNAP, while the phosphorylation or glycosylation of the bases inhibited the transcription:

<https://doi.org/10.1039/C7CC08053K>

<https://doi.org/10.1039/c9sc00205g>

<https://doi.org/10.1002/chem.202200911>

<https://doi.org/10.1039/D2CB00133K>

Some bacteriophage related modifications (including very related 5-cyanomethylU) were found to increase transcription even with bacteriophage RNAP:

<https://doi.org/10.1038/s42004-024-01354-5>

I think it is very likely that the 5cxmC will also have some influence on transcription and the paper would certainly become even more impactful if this putative role in regulation of transcription was studied (simple assay with bacterial and T7 RNAP is certainly doable). However, at the very least the authors should discuss this issue.

Reviewer #2

(Remarks to the Author)

In the manuscript entitled "Carboxymethylcytosine, a natural base modification as a handle for bacteriophage DNA hypermodification", Yang and colleagues characterize bacteriophage genes involved in DNA modification and hypermodification. By integrating several techniques, including biochemical reconstitution, X-ray crystallography, DNA sequencing, and liquid chromatography–mass spectrometry, the authors show that CmoA, CmoX, and CmoY, proteins act sequentially to achieve cytosine hypermodification in phage DNA.

First, the authors observed that bacteriophages encode a homolog of bacterial CmoA, an enzyme previously known to be

involved in 5-carboxymethoxyuridine modification pathway. Biochemical assays demonstrated that CmoA efficiently produces carboxy-S-adenosyl-L-methionine (Cx-SAM). This metabolite is then utilized by a DNA cytosine-5-methyltransferase homolog, CmoX, which introduces cytosine carboxymethylation in DNA. The authors further showed that this modification serves as a handle for an ATP-dependent ligase, CmoY, to introduce additional hypermodifications. Together, these enzymatic steps are proposed to protect bacteriophage genomes from diverse nucleic acid-targeting defense systems.

Importantly, the study demonstrates that this previously reported “unnatural” base modification occurs naturally in phage DNA, specifically at GC dinucleotide motifs. Bioinformatic analysis further revealed that homologs of these enzymes are widespread across diverse phages. Since even single amino acid mutation in nucleic acid-modifying enzymes can drastically alter the type of hypermodification, the characterization of additional phage homologs may uncover novel DNA modifications that warrant further investigation.

Overall, this is a well-designed and carefully executed study with clear results supported by multiple complementary techniques. The in vitro activities of the enzymes were validated using phage DNA analysis, thus providing the biological relevance as well. I am generally supportive of this article’s consideration for publication in Nature Communications. However, I recommend that the following issues be addressed prior to acceptance.

Main comments:

- The authors should report the percentage sequence similarity between the bacterial and phage nucleic acid-modifying enzymes—CmoA, CmoX, and CmoY.
- Figure 1a lacks clarity. It is not clear which E. coli strain is represented. The genome accession ID and genomic positions are also missing in the manuscript.
- In Figures S2, differences are observed in the catalytic activities of SpCmoA and EcCmoA. Could this be due to the MBP-tag fusion enhancing solubility and stability of SpCmoA under assay conditions?
- Figure 5c: Please provide the RMSD between the two structures.
- The authors haven’t discussed how the methodology for protein structure models shown in Figure 8d and Figure S8A?
- The Ramachandran statistics are missing in the X-ray crystallography data table. Please add.
- In several parts of the manuscript, the writing can be improved to enhance clarity for readers. For instance, in the Results section: “The genomic proximity of SpCmoY with CmoX suggests that SpCmoY may catalyze ATP-dependent ligation of the carboxymethyl group to an unidentified amine substrate. To test this hypothesis...” It is unclear how the authors arrived at this speculation. A brief statement noting that SpCmoY shares similarity with asparagine synthetase would help justify this assumption in this result section.

Reviewer #3

(Remarks to the Author)

The manuscript by Yang et al describes identification and characterisation of a phage-encoded gene cluster encoding enzymes that generate a new DNA modification. This is an exciting new addition to the growing diversity of DNA mods/hypermods.

I congratulate the authors - the manuscript was well written and satisfying to read. I found the conclusions well supported by the evidence. This is a super addition to the field and likely will be of wide interest.

My major thoughts - none of which are a requirement - is that with your wonderful expression systems more could be done on making mutants and characterising DNA binding, nucleotide selection. Also, though the manuscript says enzymes from newly identified clusters were not soluble, perhaps source the DNA and degrade it to look for other mods? Finally, as this is to do with phage defence, how about a phage assay showing this mod can protect from a defence system (or modify a plasmid and check transformation efficiency).

Minor points

Fig 1A - please add accession numbers so it is easy to find these genes

Line 140 - I don’t understand what you mean here, you are just adding a plasmid? That is not reconstitution as it’s only one component.

Supp Table 3 - please provide ramachandran favoured/allowed/outliers

Version 1:

Reviewer comments:

Reviewer #1

(Remarks to the Author)

I already liked and recommended the original version and my suggestion was not a requirement. I appreciate that the authors have at least tried the transcription study - I agree that it will be a project on itself and could be the subject of a follow up paper in the future. The remark they added is adequate.

I fully recommend the revised version for acceptance in the present form.

Reviewer #2

(Remarks to the Author)

The authors have addressed all my concerns. In my opinion, the revised manuscript is suitable for consideration for publication in Nature Communication journal.

Reviewer #3

(Remarks to the Author)

The authors have appropriately addressed my comments. No further suggestions. It looks super, congratulations!

Reviewer #1:

This is a highly interesting, though very specialized, paper on the discovery of the biochemical pathway how bacteriophage S-B43 modifies its genome by carboxymethylation of C. The suitability to Nat. Comm. is a borderline case, because it is very very specialized field (the findings concern only one single bacteriophage), but the mechanism is so interesting that I still recommend publication in Nat. Comm.

The authors have identified the carboxymethyltransferase that transfers the carboxymethyl group from carboxy-SAM to DNA - this is a novel finding of this work, although the presence of the 5cxmC in the bacteriophage genome was known before. They succeeded in determination of X-ray structure of the enzyme with carboxy-SAM which sheds light on the mechanism and specificity. Not surprisingly, the 5cxmC modification protects the DNA from cleavage by RE. Very intriguing and interesting is the discussion of the repurposing of the bacterial enzymes for modification of the virus genome.

The experiments are performed well and the discussion is relevant. The only missing and potentially very interesting results would be the functional assay how the presence of the modification influences transcription - see below. Otherwise I like the work and enthusiastically recommend acceptance subject to revision:

Response: We thank the reviewer for the positive comments.

Major point

Several previous works showed that some bacteriophage DNA modifications may not only serve as protection against RE cleavage but also as modulators or transcription. For examples, 5hmU and 5hm as well as some non-natural homologues (i.e. 5-ethylU) were found to enhance transcription by bacterial RNAP, while the phosphorylation or glycosylation of the bases inhibited the transcription:

<https://doi.org/10.1039/C7CC08053K>

<https://doi.org/10.1039/c9sc00205g>

<https://doi.org/10.1002/chem.202200911>

<https://doi.org/10.1039/D2CB00133K>

Some bacteriophage related modifications (including very related 5-cyanomethylU) were found to increase transcription even with bacteriophage RNAP:

<https://doi.org/10.1038/s42004-024-01354-5>

I think it is very likely that the 5cxmC will also have some influence on transcription and the paper would certainly become even more impactful if this putative role in regulation of transcription was studied (simple assay with bacterial and T7 RNAP is certainly doable). However, at the very least the authors should discuss this issue.

Response: We thank the reviewer for this important comment. We have conducted preliminary *in vitro* transcription experiments using 5cxmC–DNA and 5gcxmC–DNA templates with both T7 and *E. coli* RNA polymerases. In both cases, transcription was inhibited (data shown below). However, the results are not fully quantitative, as we were unable to achieve complete substitution of the modified base. The corresponding data are provided below for the reviewer’s reference only.

***In vitro* transcription of modified DNA templates by T7 and *E. coli* RNA polymerases.** **a** *In vitro* transcription by T7 RNA polymerase using DNA templates containing the T7 promoter. Templates included unmodified DNA, 5cxmC–DNA, and 5gcxmC–DNA. The resulting RNA products were analyzed by denaturing PAGE with a 96-nt spike-in control. **b** Quantification of band intensities from (a) using ImageJ. The transcription level of unmodified DNA was normalized to 1.0. **c** *In vitro* transcription by *E. coli* RNA polymerase using DNA templates containing the pveg promoter (**as described in the reference suggested by reviewer**). Although the trend of transcription inhibition by cytosine modifications was consistent with that observed for the T7 system, the signal intensity did not allow reliable quantification.

In the Discussion section, we have added that "Apart from protection against restriction endonucleases, several natural and synthetic DNA modifications have been shown to either enhance or inhibit transcription by RNA polymerases, and reactive derivatives of hydroxymethyl-dC and dU have been utilized to achieve enzymatic or photochemical control of transcription (30-34). Further studies are warranted to determine how 5cxmdC and its derivatives affect RNA polymerase activity and to explore their potential applications in transcriptional regulation."

Reviewer #2:

In the manuscript entitled “Carboxymethylcytosine, a natural base modification as a handle for bacteriophage DNA hypermodification”, Yang and colleagues characterize bacteriophage genes involved in DNA modification and hypermodification. By integrating several techniques, including biochemical reconstitution, X-ray crystallography, DNA sequencing, and liquid chromatography–mass spectrometry, the authors show that CmoA, CmoX, and CmoY, proteins act sequentially to achieve cytosine hypermodification in phage DNA.

First, the authors observed that bacteriophages encode a homolog of bacterial CmoA, an enzyme previously known to be involved in 5-carboxymethoxyuridine modification pathway. Biochemical assays demonstrated that CmoA efficiently produces carboxy-S-adenosyl-L-methionine (Cx-SAM). This metabolite is then utilized by a DNA cytosine-5-methyltransferase homolog, CmoX, which introduces cytosine carboxymethylation in DNA. The authors further showed that this modification serves as a handle for an ATP-dependent ligase, CmoY, to introduce additional hypermodifications. Together, these enzymatic steps are proposed to protect bacteriophage genomes from diverse nucleic acid–targeting defense systems.

Importantly, the study demonstrates that this previously reported “unnatural” base modification occurs naturally in phage DNA, specifically at GC dinucleotide motifs. Bioinformatic analysis further revealed that homologs of these enzymes are widespread across diverse phages. Since even single amino acid mutation in nucleic acid–modifying enzymes can drastically alter the type of hypermodification, the characterization of additional phage homologs may uncover novel DNA modifications that warrant further investigation.

Overall, this is a well-designed and carefully executed study with clear results supported by multiple complementary techniques. The in vitro activities of the enzymes were validated using phage DNA analysis, thus providing the biological relevance as well. I am generally supportive of this article’s consideration for publication in Nature Communications. However, I recommend that the following issues be addressed prior to acceptance.

Response: We thank the reviewer for the encouraging comments.

Main comments:

- The authors should report the percentage sequence similarity between the bacterial and phage nucleic acid–modifying enzymes—CmoA, CmoX, and CmoY.

Response: Sequence comparisons of these phage enzymes with their closest *E. coli* homologs are now summarized in **Supplementary Table 1**.

• Figure 1a lacks clarity. It is not clear which *E. coli* strain is represented. The genome accession ID and genomic positions are also missing in the manuscript.

Response: The *E. coli* strain, genome accession IDs, genomic positions and UniProt IDs have been added to **Fig. 1a** as suggested.

• In Figures S2, differences are observed in the catalytic activities of SpCmoA and EcCmoA. Could this be due to the MBP-tag fusion enhancing solubility and stability of SpCmoA under assay conditions?

Response: For both EcCmoA and MBP–SpCmoA, no precipitation was observed during purification or assays, suggesting that the proteins remained stably folded. The reaction progress was linear over the 12-min assay period (**Supplementary Fig. 2**), indicating that enzyme stability was not a limiting factor under the assay conditions.

• Figure 5c: Please provide the RMSD between the two structures.

Response: We have added in the **Fig. 5c** caption that “C α -based superposition of the two structures shows an RMSD of 1.027 Å.”

• The authors haven't discussed how the methodology for protein structure models shown in Figure 8d and Figure S8A?

Response: We have added in the figure captions that the structures were generated using AlphaFold 3.

• The Ramachandran statistics are missing in the X-ray crystallography data table. Please add.

Response: The Ramachandran statistics have been added to **Supplementary Table 4**.

• In several parts of the manuscript, the writing can be improved to enhance clarity for readers. For instance, in the Results section: “The genomic proximity of SpCmoY with CmoX suggests that SpCmoY may catalyze ATP-dependent ligation of the carboxymethyl group to an unidentified amine substrate. To test this hypothesis...” It is unclear how the authors arrived at this speculation. A brief statement noting that SpCmoY shares similarity with asparagine synthetase would help justify this assumption in this result section.

Response: We thank the reviewer for pointing out the need for clarification. We have added that “*SpCmoA* and *SpCmoX* are located adjacent to a gene encoding a homolog of asparagine synthetase, an ATP-dependent amide ligase, which we designate *SpCmoY*.”

Reviewer #3:

The manuscript by Yang et al describes identification and characterisation of a phage-encoded gene cluster encoding enzymes that generate a new DNA modification. This is an exciting new addition to the growing diversity of DNA mods/hypermods.

I congratulate the authors - the manuscript was well written and satisfying to read. I found the conclusions well supported by the evidence. This is a super addition to the field and likely will be of wide interest.

Response: We thank the reviewer for the positive words.

My major thoughts - none of which are a requirement - is that with your wonderful expression systems more could be done on making mutants and characterising DNA binding, nucleotide selection. Also, though the manuscript says enzymes from newly identified clusters were not soluble, perhaps source the DNA and degrade it to look for other mods? Finally, as this is to do with phage defence, how about a phage assay showing this mod can protect from a defence system (or modify a plasmid and check transformation efficiency).

Response: We thank the reviewer for these valuable suggestions.

- Indeed, enzyme engineering is part of our ongoing plan, aimed at modifying the sequence specificity of CmoX and the amine substrate preference of CmoY.
- When we initiated the project, we made considerable efforts to source suitable phages but faced difficulties due to limited strain availability and import restrictions, and were only able to obtain *Synechococcus* phage S-B43. Many of the sequences analyzed originate from uncultured phages. We are actively seeking additional isolates, particularly those containing divergent homologs of CmoY, for future studies.
- Indeed, the physiological relevance of this modification in phage defence is highly intriguing. It parallels the well-characterized hmdC and hmdU modifications, which undergo diverse glycosylations, some of which can be counteracted by bacteria [Hossain, Amer A., et al. "DNA glycosylases provide antiviral defence in prokaryotes." Nature 629.8011 (2024): 410-416]. We hypothesize that the carboxymethyl group enables similarly diverse amide modifications, and our current priority is to investigate this chemical diversity and its role in the molecular arms race.

Minor points

Fig 1A - please add accession numbers so it is easy to find these genes

Response: The UniProt accession numbers have been added to **Fig. 1a**.

Line 140 - I don't understand what you mean here, you are just adding a plasmid? That is not reconstitution as it's only one component.

Response: We have re-phrased the description to clarify: “We next attempted *in vivo* reconstitution of the CmoA-CmoX system in *E. coli* BL21(DE3) to determine whether it could modify plasmid DNA. Since *E. coli* natively expresses CmoA to produce the carboxymethyl donor Cx-SAM, we expressed only *SpcmoX*, from a modified pET28a (+)-HT vector.”

Supp Table 3 - please provide ramachandran favoured/allowed/outliers

Response: The above Ramachandran statistics have been added to **Supplementary Table**.